# Resource sharing in technologically defined social networks

Hirokazu Shirado[1,2], George Iosifidis[3,4], Leandros Tassiulas[1,5] & Nicholas A. Christakis [1,2,6]

Technologically enabled sharing-economy networks are changing the way humans trade and collaborate. Here, using a novel 'Wi-Fi sharing' game, we explored determinants of human sharing strategy. Subjects ($N = 1,950$) participated in a networked game in which they could choose how to allocate a limited, but personally not usable, resource (representing unused Wi-Fi bandwidth) to immediate network neighbors. We first embedded $N = 600$ subjects into 30 networks, experimentally manipulating the range over which subjects could connect. We find that denser networks decrease any wealth inequality, but that this effect saturates. Individuals' benefit is shaped by their network position, with having many partners who in turn have few partners being especially beneficial. We propose a new, simplified "sharing centrality" metric for quantifying this. Further experiments ($N = 1,200$) confirm the robustness of the effect of network structure on sharing behavior. Our findings suggest the possibility of interventions to help more evenly distribute shared resources over networks.

[1] Yale Institute for Network Science, Yale University, New Haven, CT 06520, USA. [2] Department of Sociology, Yale University, New Haven, CT 06520, USA. [3] School of Computer Science and Statistics, Trinity College Dublin, Dublin 2, Ireland. [4] SFI Research Centre CONNECT, Dublin 2, Ireland. [5] Department of Electrical Engineering, Yale University, New Haven, CT 06520, USA. [6] Department of Biomedical Engineering, Yale University, New Haven, CT 06520, USA. Correspondence and requests for materials should be addressed to N.A.C. (email: nicholas.christakis@yale.edu)

Resource sharing is rapidly gaining renewed significance in economic and social life, and collaborative consumption[1], or the sharing economy[2], has attracted great interest. These novel types of sharing solutions to socioeconomic challenges have the potential to substantially improve the utilization efficiency of limited resources by reallocating them to those in need in ways that are relatively costless to the sharer, and in ways that set the stage for reciprocation by the recipient. The recent growth of this disruptive economic model has been spurred by internet connectivity and the proliferation of mobile computing and online social networking platforms[3–6]. These innovations allow people to connect in technologically mediated social networks and exchange resources in a peer-to-peer fashion[7–9]. Indeed, it is expected that individuals and local communities will organically play a key role in these decentralized sharing systems[2].

Human cooperation and resource sharing have been studied in different contexts and with diverse methods, from observing hunter-gatherer societies[10,11], to conducting behavioral experiments involving public goods games[12–16], to exploring theoretical models for cooperation equilibriums[17–19]. The emergence of cooperation generally depends on how humans respond to their social surroundings; cooperative partners generally induce cooperative behavior[14,20,21], and the number and structural arrangements of available partners also matters[22,23].

However, sharing-economy models include many novel features, which have been relatively unexplored. For instance, humans are both resource consumers and producers (or, prosumers), and these roles change quickly as people make repetitive sharing decisions[2]. These technology-mediated sharing schemes are nowadays instantiated on an unprecedented scale with respect to their volume, numbers of participants, and spatiotemporal granularity of the shared resources. Typically, participants are able to decide and employ a different sharing strategy for each of their potential collaborators. These special features might affect the decisions of humans who, for example, could attempt to strategically disperse their limited resources to others, in anticipation of larger reciprocated shares in the future.

Furthermore, modern sharing models contain an underlying network that prescribes humans' sharing opportunities. For example, ride sharing or food sharing is constrained by the geographic proximity of the participants (spatial network constraints[3]); renewable energy sharing relies on the grid network (technological network constraints[4]); and peer-to-peer resource exchanges are conditioned on the matching of the users' needs (preference constraints[5]).

The impact of networks on our social and economic interactions has been experimentally validated with respect to cooperation, and networks may have significant impact on modern sharing interactions as well. For instance, the network might increase the collaboration opportunities for some participants and impede those of others, or amplify sharing inefficiencies in other ways. Understanding features of socio-technical networks that facilitate sharing is important in establishing large-scale and successful sharing systems. Yet, there are few studies, if any, to clarify the combined effect of technical specifications governing social interactions and the behavior of humans engaged in resource sharing over networks.

Here, we perform social network experiments which encompass the salient features of modern sharing-economy models[24]. Specifically, we examine individuals' sharing strategy and the aggregate outcome of humans' interactions at the network level. Furthermore, we study topological and behavioral changes associated with the manipulation of the extent of connectivity possible in the network as governed by the supposed range of devices that specify the connectivity. We also explore how sharing inequality arises and how it can be reduced through interventions on the network structure. That is, we investigate what types of technical and social networks facilitate equal (or unequal) distributions of resources.

In order to examine the sharing behavior of the subjects, we developed a novel sharing-economy game based on the notion of a household Wi-Fi sharing service ("Wi-Fi sharing game"). There is an increasing interest in the telecommunication industry in mechanisms that enable users to share their Internet connections or other types of network resources (e.g., node computing or storage capacity)[24,25]. Although prior work in engineering has proposed technical solutions for enabling such sharing, this work neglects considerations arising from actual human behaviors[24,26]. This sharing game, which simulates real-world applications[25], allows us to examine realistic sharing dynamics incorporating actual human interactions. While framed as a game involving the sharing of Wi-Fi over geographic distance, our setup has a number of generic features that are applicable to many settings where people have machine-mediated and technologically defined social interactions and share a resource.

We recruited 600 human subjects via the online labor market Amazon Mechanical Turk, and we randomly assigned them to one of three conditions in a series of 30 sessions (10 sessions per condition). Subjects were randomly assigned to a location in a network of 20 nodes. The networks were generated using a random geometric graph model[27] that qualitatively captures the spatial deployment of Wi-Fi networks (Fig. 1). The 20 nodes were placed uniformly at random in a unit square, and two nodes were connected if their Euclidean distance was smaller than a certain connection radius. This latter parameter can capture the impact of a communication technology (here, the coverage area of the Wi-Fi routers) which, in general, may depend on a network device's capabilities (e.g., the range of the routers) and is manipulable by central policy-makers (e.g., the manufacturer or the Internet service provider).

Subjects then played this Wi-Fi sharing game for 15 rounds (per session), without knowing each session's duration in advance. Each subject was given a certain amount of Wi-Fi capacity, 30 units per round, which he or she did not need during sequential time intervals (e.g., simulating an absence from his or her residence). The subject could allocate part of this excess resource (which was useless to them) to each one of her/his neighbors during these idle hours, conditioned on the connection range of the Wi-Fi router (which, inter alia, specifies the density of the social network they are embedded in, from which they could choose sharing partners). Albeit hypothetical, such fine-grained sharing decisions are technically possible today and commercially available[24].

While making their decisions, subjects were only given the information regarding their own resources and transactions. They were informed neither about the exchanges of their neighbors with the other players, nor about the amount of resource their neighbors could share with them (see Methods). Subjects were allowed to give their neighbors different amounts of the resource, possibly recognizing past exchanges of their own or anticipating future ones, and they did not have to allocate their entire capacity. The non-allocated capacity neither carried over to the next round nor counted towards their score (hence, it was a wasted resource). Each subject's final score depended only on the units received from neighbors, which were regarded the subject's wealth in the game and converted to actual monetary compensation at the end (see Methods). The goal of the game was to collect as much of this as possible over the course of the game.

In this experiment, the individual's choice was not whether to cooperate or not, as in other social-dilemma situations, but rather with whom and how much to cooperate (thus coming to define the actual social ties from among those made possible by the

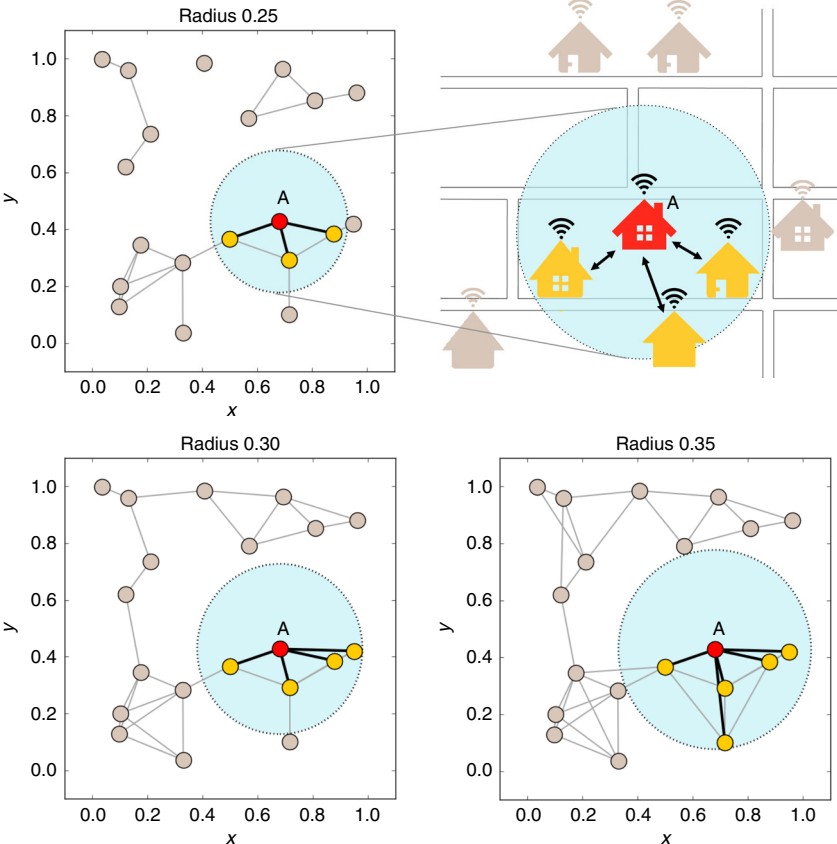

**Fig. 1** Sharing game with different connection ranges. The game simulates a household Wi-Fi sharing service as an example of sharing an otherwise unusable resource. Subjects can share their resource (Internet access) with neighboring households within a particular geographic range. We generated network structures using a random geometric graph model. The model places 20 nodes uniformly at random in the unit square; two nodes are joined by an edge if the Euclidean distance between the nodes is within the specified radius. We manipulated the connection radius with three levels: 0.25, 0.30, and 0.35. By increasing the radius, as shown by the light blue circles, the network density increases; for example, the player placed at node A increases the number of neighbors with whom she or he can potentially share the resource (thus forming an actual social tie, based on the underlying technological network)

underlying technological system)[26,28]. These decisions about with whom and how much to cooperate were coupled due to the limited resource at the disposal of each subject at each round[29]. Therefore, the Wi-Fi sharing game has a competitive structure that is not present in other cooperation games. On the other hand, it also differs from other games, such as the Prisoners Dilemma, since the ego's actions do not directly affect her own utility, and moreover there is coupling among the ego's sharing decisions towards the different alters[30]. Supplementary Figure 1 summarizes these characteristics.

Within this basic setup, we manipulated the connection radius of network formation and hence the underlying network structure, with the following three values (based on unit square dimensions): 0.25, 0.30, and 0.35 (Fig. 1). This experimental condition represents the impact that an actual engineering intervention might consequently have, namely (in this case) an increase of the access range of the Wi-Fi routers in the game scenario (e.g., by improving the protocol it employs, or the antenna gain). Wider connection ranges increased the exchange options for each subject. That is, the underlying network density (the fraction of ties present in the network versus the number of all possible ties) increases monotonically with the connection range. While the geographical relative position remained the same (i.e., the subjects did not relocate), the geodesic structure of the possible interaction network varied with the technological manipulation. We randomly generated networks with 10 different

X–Y coordinate settings of 20 nodes with each of the three connection radii. Subjects were randomly assigned to one of the 30 networks (10 network settings by three connection radii) and not allowed to participate in more than one session.

We find that the resulting network structure affects the variance in acquisition of the shared resource, and that denser networks decrease any wealth inequality, but that this effect diminishes, revealing a saturation property. Individuals' benefit is shaped by their network position, and the benefit increases if they have a particular kind of structural leverage, namely, if they have many partners who in turn have few partners. We propose a new, simplified "sharing centrality" metric for quantifying this. Further experiments confirm that the impact of network structure on sharing behavior is robust to revealing more information about partners (such as the number of their trading partners or the resources they receive). Our findings suggest the possibility of network interventions to help more evenly distribute shared resources by controlling network structure using technological specifications.

## Results

**Effect of connection range on wealth distribution**. Variation of the connectivity range affected the realized network structure and the cumulative wealth inequality (see Fig. 2 for network snapshots, including of both the underlying technologically defined

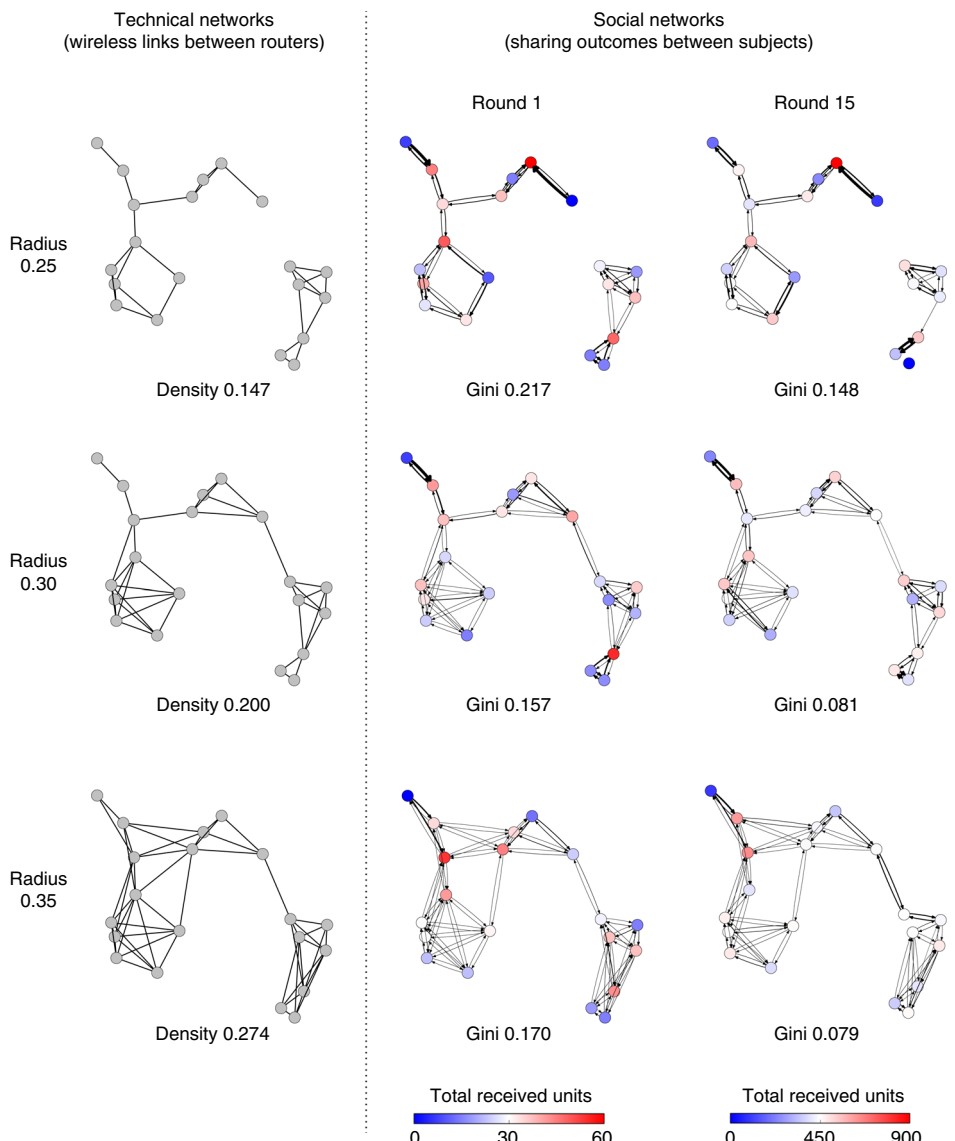

**Fig. 2** Network snapshots and actual realized exchanges of a resource with different connection ranges. The network samples have the same geometric configuration (i.e., location of the nodes on the plane) but different connection ranges. In the social networks, node color indicates total received units (with redder nodes being richer; with bluer nodes being poorer). Arrow width indicates total given units from sender to receiver (with line width indicating more giving to the neighbor indicated with the arrowhead). In addition, the graphs exclude extra-thin arrows according to the threshold that the given units are less than 10% of total resources. Both properties are normalized with the number of rounds. As the session progressed, the players selected sharing partners to seek mutual exchange. As a result, the wealth gap decreased

network and the actual, realized social network, based on sharing interactions). The average subject's wealth (i.e., the total units subjects received during the game) did not vary with the connection range (see Supplementary Figure 2), but the Gini coefficient[31] of the subjects' score decreased noticeably as the connectivity range increased, i.e., there was more even sharing as the range of the "device" and network density was increased (Fig. 3). However, the change in wealth inequality is diminishing with the increase in network density. That is, while the Gini coefficient decreased by 0.074 from the radius of 0.25 to 0.30 ($P <$ 0.01; paired $t$-test with $N = 10$ sessions), it decreased by 0.022 from the radius of 0.30 to 0.35 ($P = 0.11$; paired $t$-test with $N = 10$ sessions). We also confirmed the non-linearity between network density and sharing inequality with a quadratic regression model (see Supplementary Table 1). This suggests a threshold property in terms of the possible reduction of sharing inequality and the necessary cost for achieving it.

The final Gini coefficients, which are shown in Fig. 3, result from a combination of the underlying network structure and the observed human behaviors. To clarify the structural effect separately from the behavioral one, we introduce a null model that shuffles the receivers to which subjects gave the resources while keeping the diversity of the observed allocations from the actual experiment. Figure 4 shows the comparison of the actual dynamics of the Gini coefficients with that of the random-shuffle simulations with 1000 repetitions. At the beginning of a session, the actual Gini coefficients are almost identical to the simulation results that represent the structural impact on wealth inequality. As the session progresses, the actual wealth inequality among subjects decreases to a steady level, while the structural impact does not change. Individuals' resource allocation strategy causes the improvement in wealth inequality over the null model. On the other hand, in all cases, the subjects do not reach the most efficient sharing equilibrium, although, in theory, a zero Gini

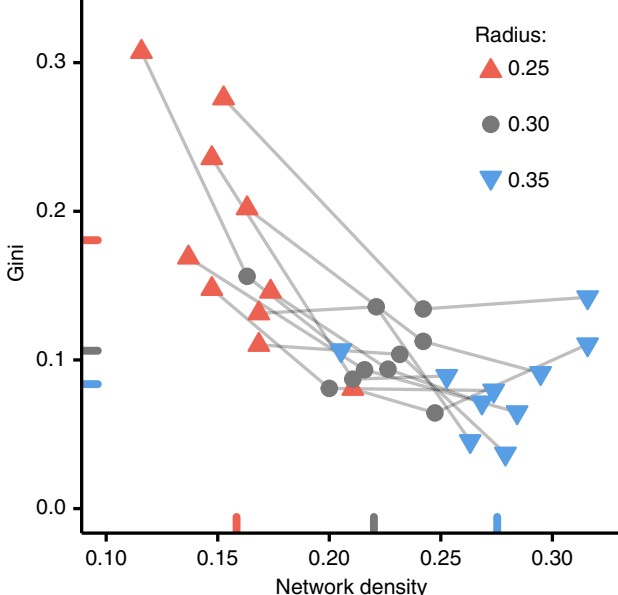

**Fig. 3** Wider connection range decreases economic inequality, but the change is diminishing. Dots indicate network density and final Gini coefficient for each session ($N = 30$ sessions). Light gray lines bind sessions having the same geometric configuration (i.e., location of the nodes on the plane) set with different connection ranges (e.g., the networks of Fig. 2) shown as dot color and shape. Average network density and Gini coefficient across the connection radii are shown as colored ticks on X- and Y-axis

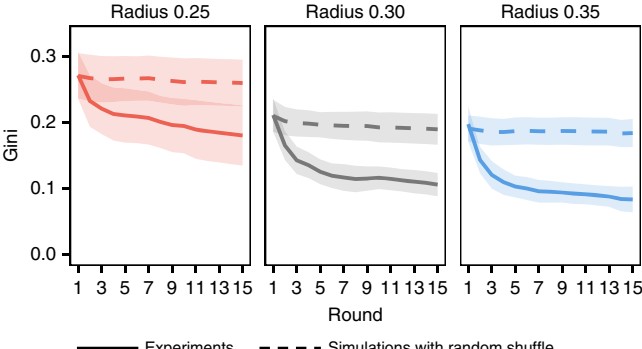

**Fig. 4** Changes in Gini coefficient across rounds. The solid lines show the average of experimental results; the dashed lines show the expected value when each individual randomly allocates their resources while preserving the diversity of the allocation from the experiment. Shaded areas denote 95% confidence intervals ($N = 10$). The actual subjects reduced wealth inequality through their interactions, compared to identical circumstances where their resources were randomly allocated

value is attainable for the tested graphs[32]. The remaining wealth inequality is highly correlated with the hypothetical one reflecting only network structure (Pearson correlation coefficient = 0.872; $P < 0.01$ with $N = 30$ sessions). This result demonstrates that not only human behavior but also network structure affect wealth inequality in sharing. We find similar results with another null model where subjects engage in equal sharing with their neighbors (see Supplementary Figure 3).

**Individual reciprocity given network constraints**. Next, we explore individuals' resource allocation strategy and the effect this has on wealth inequality. We find that subjects are likely to seek

reciprocity in their transactions (Fig. 5a). The individual-level analysis shows that, on average, subjects increase their allocation to the neighbors who had given more than they received at the previous round, and the opposite is also true. The estimated coefficient for the impact of the last-round sharing balance on the next-round increase (or decrease) in resource sharing to the neighbor is significantly positive for all the connection radius treatments ($P < 0.01$ for all the connection radius treatments; regression analysis with random effects for rounds and individuals; see Supplementary Table 2). The reciprocal tendency is also statistically significant across all network degrees (number of neighbors) of subjects.

Subjects increase their allocation to their generous neighbors (i.e., those from whom they receive more resources than what they have offered to them in the previous round) and decrease it to their stingy neighbors. Note that each individual has only 30 units, and hence fully reciprocated interactions are not always possible. Despite this limitation, however, our findings are consistent with a reciprocal strategy[29,33–36] instead of an exploitative strategy[37], where the latter would have led subjects even to reduce the allocations towards the neighbors who already respond with high sharing (hence needed no further incentive to collaborate). While direct reciprocity increases with increasing number of neighbors per subject (i.e., network degree) up to a degree of 3 (i.e., when subjects have three neighbors), the impact of network degree appears to diminish when subjects have more than three neighbors in this game setting (see Supplementary Figure 4).

As a result, on a network level, in general, subjects develop symmetric, reciprocal exchanges where they exchange equal amounts of resources (Fig. 5b). The network-level reciprocity[38] reaches significantly higher values compared to hypothetical expectations generated by the random-shuffle model ($P < 0.01$ for all the connection radius treatments; paired $t$-test with $N = 10$ sessions at round 15). The individuals' reciprocal behavior reduces the gap between given and received units in dyads to some extent, and so the wealth inequality among subjects decreases over the rounds (Fig. 4).

Figure 5 also shows how individual-level and network-level reciprocity varied in different network graphs. The networks with a small radius of 0.25 show a smaller impact of the last-received on the next-given resource amounts, compared to the networks with radius 0.30 and 0.35 (Fig. 5a). As a result, subjects in the networks with radius 0.25 reached a lower level of reciprocity over the rounds, compared to those in the networks with radius 0.30 and 0.35 (Fig. 5b). As the random assignment ensures no difference in subjects' distribution with respect to their innate sharing propensity, this difficulty in carrying out the reciprocal allocation is due to the network topology.

**Centrality metrics for network sharing**. The network-level analysis shows structural as well as behavioral effects on sharing inequality. The individual-level analysis also suggests that, while subjects show high reciprocity in their behavior, the resource sharing benefits differed among individuals due to their network position. Prior studies suggested that the probability of a mutually-agreed exchange between connected nodes is an important structural feature of the geodesic location of a node, and that this affects its expected benefits in bargaining networks[37]. The probability was termed graph-theoretic power index or GPI (the original paper called the measure "GPI3," but we simply call it GPI here). We measured GPI with respect to sharing networks and studied whether it is correlated with the benefit each node accumulates (Fig. 6a; see the caption of Fig. 6 for measurement details pertaining to GPI). We also compared the predictive power of the GPI metric with other node centrality metrics, including a simplified one, which we term "sharing centrality," that we introduce here.

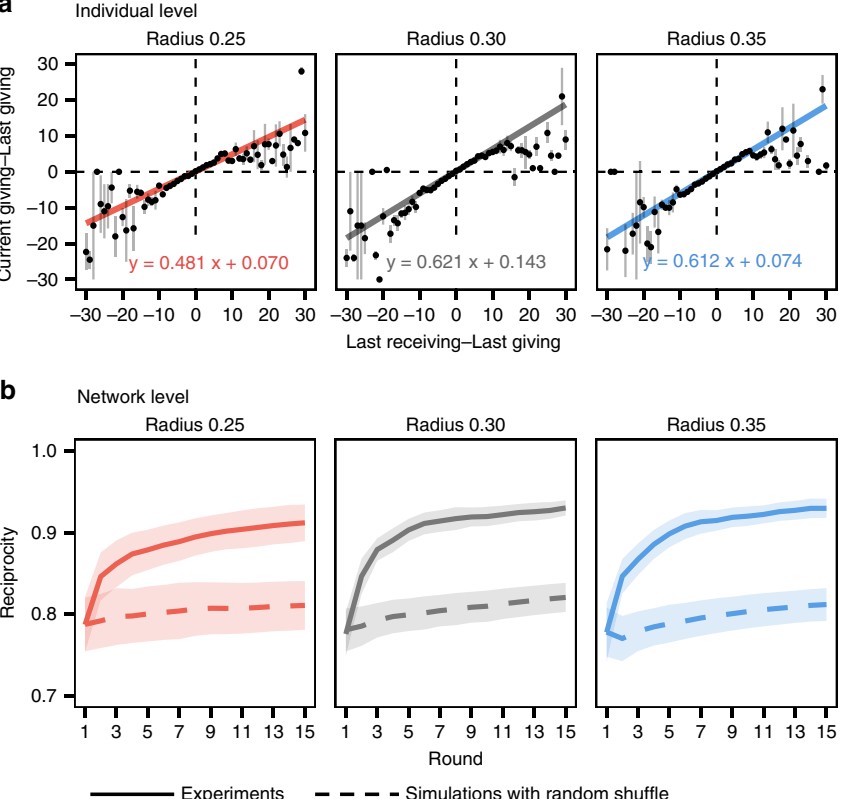

**Fig. 5** Subjects reciprocate to their neighbors. **a** Individual-level reciprocity. The graphs show how much a subject increases (or decreases) their share based on the last transaction balance they had with each one of their neighbors. Dots indicate the average of actual data. Error bars are standard errors. Colored lines are the estimated result of regression analysis with nested random effects for round and individuals. On average, subjects raise their allocations when their neighbor were generous, i.e., gave more than they had received from the egos (so had positive values on the X-axis), as indicated by the positive slopes of the regression line ($P < 0.01$ for all the radius treatments). The reciprocal tendency is smaller in the network of radius 0.25, compared to in that of radius 0.30 or 0.35. **b** Changes in network-level reciprocity across rounds. The reciprocity of the weighted networks is calculated by dividing the total reciprocated resources (i.e., the sum of symmetric sharing) by the total shared resources in the network. The solid lines show the average of experimental results; the dashed lines show the expected value when each individual randomly allocates their resources while keeping the diversity of the observed allocation from the experiments. Shaded areas denote 95% confidence intervals ($N = 10$)

We find that GPI is highly correlated with the total received units of each subject in the game (the correlation coefficient is 0.70, $P < 0.01$; Pearson correlation test; Fig. 6b). This structural feature of the geodesic location of a subject largely determines a subject's wealth in a sharing network. On the other hand, other conventional network centrality measures, such as degree centrality (as noted above), betweenness centrality[39], or eigen-vector centrality[40] are less associated with wealth accumulation (Fig. 6d). We also test Bonacich power centrality with several negative beta values[41], and find that it is less predictive than GPI for individual wealth (see Supplementary Figure 5). Furthermore, the correlations between GPI and wealth were statistically the same across the connection radii (see Supplementary Figure 6). We also tested these centralities using regression models incorporating a random effect for sessions and find similar results (see Supplementary Table 3).

Why is individual wealth determined by GPI rather than network degree (or the other network structural attributes)? In contrast to information flow, where being connected to well-connected people is helpful, individuals are more likely to reap a benefit in holding a monopolistic position when it comes to resource sharing. That is, in situations like this, not only must they have many neighbors (high degree of ego) but also their neighbors should have few neighbors (low degree of alter)[37]. This effect continues to further geodesic distances. For example, if your

neighbor's neighbor has many neighbors, your neighbor is unlikely to share many resources with him or her; thus, you are likely to receive many resources from your neighbor because of the high degree of alter's alter.

Motivated by this observation, we introduce a new type of node centrality that we name "sharing centrality" and define it as the sum of the reciprocal degrees of the one-hop neighbors of the ego (Fig. 6a). We find that this metric predicts the individual wealth significantly better than the degree centrality and at least as well as the GPI metric (Fig. 6c). But, in contrast to GPI, the sharing centrality requires only local information. This light computational load is essential for interventions in large-scale and dynamic sharing networks. In modern sharing circumstances where millions of people continue to move in and out, previous global knowledge of the entire network structure, which GPI requires, is impractical. In addition, sharing centrality can clarify how much network structure itself affects individual wealth through sharing interactions. If individuals randomly responded to their neighbors (i.e., the random-shuffle model), sharing centrality would be perfectly correlated to individual wealth (the impact without actual human behaviors is obscure in GPI; see Supplementary Figure 7).

**Degree assortativity and wealth inequality**. In light of this flip-flop impact of network degree, degree-disassortative networks[42]

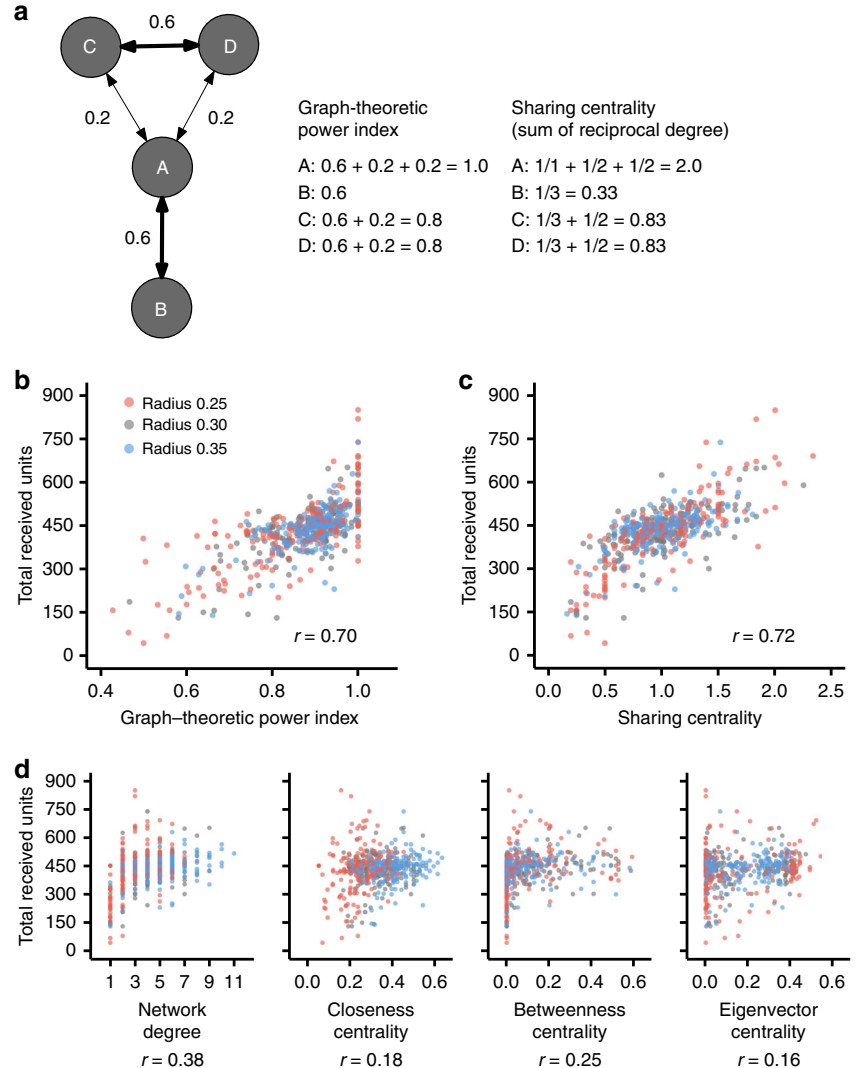

**Fig. 6** Power index and sharing centrality are highly correlated with the economic benefit in a sharing network. **a** Graph-theoretic power index (GPI) captures the probability the focal node will be engaged in a successful mutual selection with one of his or her alters. This in turn, gives the node a structural advantage in bargaining in strategic interactions. The easiest way to understand how the geodesic position benefits a node is the following simple scenario. Suppose node A randomly selects one of her neighbors (say node C) in order to allocate all its resource; then if C has also selected node A as its unique collaborator, asymmetric and fully reciprocal relationship can be established and both node C and A will stop seeking collaborators. Otherwise, node A can proceed and select another node as a potential collaborator. This iterative process continues until a mutual selection is achieved for every possible pair of nodes. It is easy to see that each dyad has a different probability of mutual selection throughout this iterative process. Namely, the probability of mutual selection (by concurrent nomination in an repetitive process) is 0.6 for the edges **a, b** and **c, d**, and 0.2 for **a–c** and **a–d**. In total, this means that node A can find a full reciprocator with probability 1, while for node B the probability is 0.6 and for nodes C and D it is 0.8. Sharing centrality is the sum of reciprocal network degree among alters. The centrality approximates the structural advantage, which GPI represents, but using only local topology information. **b–d** Dots indicate several measures of network centrality for each subject and his or her total received units at the end of game (out of $N = 600$ subjects who participated). Dot color indicates the network's connection radii; red for radius = 0.25, gray for radius = 0.30, and blue for radius = 0.35. GPI and sharing centrality show clear correlation with their economic benefit from a sharing game (**b**, **c**). This correlation is higher than all the other centrality measures (**d**). The "$r$" values indicate Pearson correlation coefficients ($P < 0.01$ for all the correlation coefficients)

are likely to have large variance of sharing centrality and, as a result, large wealth inequality in resource sharing, compared to degree-assortative networks. We tested this hypothesis with another, separate experiment using two simple networks (Table 1) ($N = 150$ subjects in 10 sessions). The networks have the exact same degree distribution, but opposite degree assortativity. Since the degree distribution was the same, subjects played the game in identical local conditions. Nevertheless, the network having negative degree assortativity, with large variance of sharing centrality, generated high wealth inequality, compared to that of positive degree assortativity (Table 1). That is, regardless of

network density and degree distribution, sharing centrality alone appears to influence wealth inequality in sharing networks.

**Robustness of the network effect on sharing behavior.** Last, in separate, further set of experiments involving 1200 subjects, we find that the structural impact on wealth inequality is robust even when subjects obtain additional information about their local neighbors' status. In the "wealth-visible" condition, subjects could observe, in addition to the previous-round transactions, the total units previously received by each of their direct neighbors. In the "degree-visible" condition, subjects could additionally see the

**Table. 1 Economic inequality depending on degree assortativity**

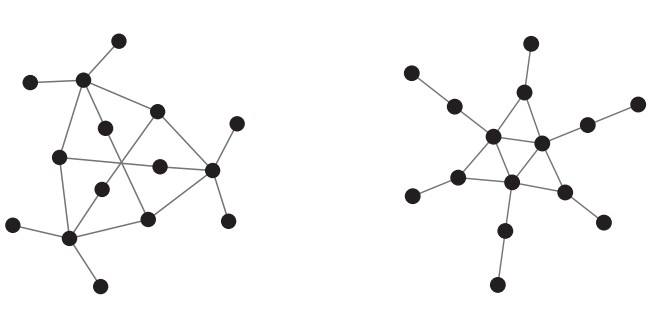

|  | Network A | Network B |
|---|---|---|
| *Network structure* |  |  |
| Number of nodes |  |  |
| Degree = 1 | 6 | 6 |
| Degree = 2 | 3 | 3 |
| Degree = 3 | 3 | 3 |
| Degree = 5 | 3 | 3 |
| Network density | 0.171 | 0.171 |
| Degree assortativity | −0.744 | 0.233 |
| Sharing centrality |  |  |
| Degree = 1 | 0.2 | 0.417 |
| Degree = 2 | 0.533 | 1.2 |
| Degree = 3 | 0.9 | 1.4 |
| Degree = 5 | 3.167 | 1.567 |
| S.D. | 1.114 | 0.493 |
| *Experiment result* |  |  |
| Total received units |  |  |
| Degree = 1 | 130.8 | 270 |
| Degree = 2 | 355 | 427 |
| Degree = 3 | 443.9 | 517.1 |
| Degree = 5 | 838.3 | 475.8 |
| Gini coefficient | 0.413 | 0.194 |

Networks A and B have the same degree distribution but different degree assortativity and hence distributions of sharing centrality. The experimentally observed Gini coefficient (calculated by total received units of the 150 subjects after 15 rounds) in Network A is more than twice as large than as that in the Network B in the sharing setting ($N = 5$ sessions per network)

number of connections (network degree) of their neighbors. The objective here was to test if this information changed how subjects assessed their neighbors' sharing; for instance, a neighbor who shared little but had high degree can be seen to have many conflicting demands for sharing and therefore might be tolerated and treated more generously, or a neighbor who already receives many resources from their partners (a wealthy neighbor), might be seen as one who does not need additional resources.

The statistical analysis including the above two visible conditions shows that the additional information had no impact on wealth inequality in a sharing network (see Supplementary Table 1). Moreover, GPI substantially determines individual wealth in both conditions (see Supplementary Figure 8). However, the correlation between GPI and wealth in the wealth-visible sessions is significantly smaller, compared to that in the baseline and degree-visible condition ($P < 0.01$; person correlation test), which reveals that this information weakens the effect of the network structural advantage. Similar effects have been observed in social-dilemma experiments[43], but here the impact of providing this information is smaller and not enough to fundamentally change individual-level reciprocity (see Supplementary Figure 9) and group-level wealth inequality (see Supplementary Table 1). Our sharing centrality measure yields similar results (see Supplementary Figure 10). In short, when it comes to sharing,

people here act according to who they are connected to, and how much those people have shared with them, and are less influenced by how wealthy or connected their partners are.

## Discussion

We find that manipulable technological changes can decrease social inequality by affecting network density. While reciprocal resource allocation in individuals reduces overall wealth inequality, network structure still affects each individual's sharing outcomes. We identify the cumulative probability of forming a mutual exchange as a key structural determinant of individual advantage in a sharing network. And we propose a new, simpler sharing centrality metric to quantify this property. We also confirm that the sharing dynamics are less affected by information about partners' wealth and options to share with others.

As for individual behavior, we find that subjects form largely reciprocal relations, ones where they adjust their actions in an attempt to exchange equal amounts. This finding is in line with previous studies revealing inherent reciprocity in social exchanges, even when this might result in utility loss[29,33–36]. Indeed, this is also the case here, as oftentimes non-reciprocal or asymmetric relationships might have resulted in larger resource accumulations for some players[32]. To a certain extent, this behavior might also be attributed to the specifically social value that humans attach to trading with their peers[44]. Prior work in wireless networks assumes either that humans will comply with centrally designed protocols (hence not factoring the human aspect)[24] or that they will be considered fully-rational agents (which is also not reasonable)[26,32,45]. Our results suggest that these models might not capture actual sharing dynamics because they ignore the social value of the resource sharing process.

Our game differs from other well-studied social-dilemma scenarios[17] in many respects: every player has to decide independently whether they will collaborate with each of their neighbors; the collaboration involves no cost; ego's decisions about allocations to several neighbors are coupled due to the limited resource that needs to be shared among them; and nodes with common neighbors compete indirectly for their resources. As these features encompass the basic aspects of decentralized sharing-economy models[24], we believe our findings extend to other types of exchange of finite resources or services including the sharing of fixed assets (like cars), sharing of time[46], allocation of budget or attention[28], and exchange of finite favors[35]. They also give practical insights regarding technological applications intended to enable them.

For example, our experiments show that network design features affect the outcome of sharing interactions and suggest ways to intervene in order to enhance sharing beyond manipulating the radius of social interactions. For instance, by manipulating the sharing centrality (or other topological features[47,48]) with the addition (or elimination) of links between certain nodes, the inequalities stemming from network structure might be reduced, or the amount of sharing might be increased. To extend this example, social sharing may be greatly facilitated through the thoughtful provision of extra resources in the form of public goods. For example, a few Wi-Fi hot spots provided by a central authority and strategically positioned (so as, for instance, to reduce disassortativity) might facilitate broader sharing by all members of a network, creating cascades of benefit. Our result suggests that, although one simply could place the Wi-Fi spots in locations with the greatest number of people (i.e., increase the degree of high-degree nodes), this might not reduce the sharing gap because it does not take into consideration the flip-flop impact of network degree. Another potential intervention would be to restrict or cut certain connections between parties (e.g., by controlling sharing options that users are given), even when they

are within range, which paradoxically may actually enhance overall sharing[49], or targeting participants based on their sharing centrality and teaching or nudging them to adopt more effective allocations to achieve reciprocity, which may lead to an even distribution of resources more quickly[50]. Our experiment sets the basis for further exploration of how to address disparities in a community-oriented sharing-economy, including applications unconstrained by geography.

There are features potentially relevant to inequality in sharing transactions that our experiments do not explore, for example, how peer sanctions or institutions might affect the outcome[51]. One could also explore situations where (i) the resource could be kept by the ego and used later (this would make the sharing decision harder); or (ii) there was a punishment/tax for wasting resources; or (iii) the players could transfer the resources they received from their neighbors to others; or (iv) there was a significant cost for sharing resources. Another promising topic is the effect of peer reputations on sharing behaviors[6,52] (indirect reciprocity), in keeping with notions of costly signalling[11].

The sharing economy is not a new idea. For example, hunter-gathers enact their social allegiances and secure a more regular diet through the institutionalized sharing of food[10]. From a historical perspective, sharing networks today may be a reflection of human nature[35] and a revival of ancient customs[53]—albeit driven by new, sophisticated information technology. The evidence presented here suggests that technological manipulations might induce greater reciprocation in resource sharing. Given that exchange is a foundation of human society[54], our results may help not only to design an equitable sharing service, but also to address fundamental challenges with respect to our collective well-being.

## Methods

**Experiment setup**. A total of 1800 unique subjects (plus a further 150 for the test of degree assortativity (Table 1)) participated in our incentivized economic game experiments. They were recruited using Amazon Mechanical Turk (AMT)[55,56], and they interacted anonymously over the Internet using customized software playable in a browser window (available at http://breadboard.yale.edu). Each session had 20 subjects at the outset. The subjects repeatedly interacted with their connected neighbors through a sharing-economy "Wi-Fi sharing game" that we developed, over 15 rounds. We completed 10 sessions for each treatment combination of the three levels of connection range (in networks generated with geographic graphs (with ranges of 0.25, 0.30, and 0.35 in a unit square)) crossed with three different conditions regarding neighbors' information (invisible, wealth-visible, and degree-visible). In all, 90 sessions were conducted from March to September 2016. In each session (after passing various tutorials), the subjects were paid a $2.00 show-up fee and $2.00 payoff for game completion; in addition, each subject's final score (i.e., total units received from their neighbors) summed over all the rounds was converted into dollars at an exchange rate of $1.00 = 200 units. All the subjects were informed about the use of their behavioral data for research purposes upon enrollment in the experiment (see Supplementary Methods). This research was approved by the Yale University Committee of the Use of Human Subjects.

**Information availability**. The players had access only to the necessary information for the game. First, the players were not informed of the actual number of rounds since this might have affected their sharing strategy towards the end of the game. Second, they were not informed of the overall network structure, i.e., they did not have information about the geodesic locations of all nodes in the network. Each player was able to see and interact only with their immediate neighbors. They were only able to observe the points they received from their neighbors (see Supplementary Methods for the actual game view).

In some experiments, we manipulated the players' information in the additional information-visibility scenario. In the degree-visible condition, the players were additionally informed about the degree (number of neighbors) of each of their neighbors. In the wealth-visible condition, they were additionally informed about the total score of their neighbors (but not about their degree, i.e., how many neighbors they had). When a player was assigned to either information-visible treatment, they were able to see the additional information of each neighbor in the network diagram and in the information table on their game screen (see Supplementary Methods for the actual game view).

We confirmed using subjects' IP address that all the pairs in the game networks came from different locations. That is, subjects could not share their screen physically.

**Network formation**. In the game, subjects were assigned to a location in a network of 20 nodes. The networks were generated using a random geometric graph model[27] that qualitatively captures the spatial development of the Wi-Fi networks in many residential areas. The 20 nodes were placed uniformly at random in a unit square, and two nodes were connected if their Euclidean distance was smaller than a certain connection radius (Fig. 1). We manipulated the connection radius of network formation using the following procedure. First, we created 30 networks with 0.30 as the connection radius (10 networks for each information-visible setting: base, wealth-visible, and degree-visible conditions). In this process, we excluded networks having isolated nodes in order to compare players' sharing inequality between the different information visibilities with the same network size. Second, we created networks with 0.25 and 0.35 as the connection radius using the same node locations in a unit square (i.e., X–Y coordinate settings of nodes) as the first set of networks with 0.30 as the connection radius. That is, while the geographical relative position remained the same, the geodesic structure of the interaction network varied with the connection distance.

**Players dropping during the game**. At any point during the game, if a player was inactive for 45 s, they were warned about being dropped. If they still remained inactive after 45 s, they were dropped. In some instances, additional players had to be dropped (cascade drops), namely the ones who had degree 1 and their only connection was the player who was dropped. As too many dropped players destructively affect the network structure and the sharing dynamics of the remaining players, we did not use the sessions where more than five players were dropped during the game (i.e., a retention rate ≤75%). Overall, 4 players dropped in two sessions; three players dropped in 11 sessions; two players dropped in 21 sessions; one player dropped in 25 sessions; and no player dropped in 31 sessions. Since the players who dropped in the game changed the network structure, we calculated all the node metrics in each network of 15 rounds and used the average for each subject. The dropped players were prohibited from joining another session of this experiment.

**Additional test of degree assortativity**. In addition to the main experiments (with $N = 1800$), we recruited 150 unique subjects from AMT and tested the effect of degree assortativity on sharing inequality (Table 1). We used two specially designed networks (see the network snapshots in Table 1). Both networks consist of 15 nodes and 18 edges. Both have the same degree distribution: six nodes of 1 degree, three nodes of 2 degree, three nodes of 3 degree, and three nodes of 5 degree. Each network had 15 players. However, the two networks have different connections between low-degree and high-degree nodes: one network has negative degree assortativity ($-0.744$) and the other has positive degree assortativity ($0.233$). We conducted five sessions for each treatment, and each session had 15 rounds.

## Data availability

The experimental data, all the figures that have associated raw data, and the programming codes are stored at http://humannaturelab.net/publications/resource-sharing-in-technologically-defined-social-networks at the Human Nature Lab Data Archive.

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

## Acknowledgements

We thank Forrest W. Crawford, Dimitrios Katsaros, Andrew V. Papachristos, Yongren Shi, and Martin Wainstein for helpful comments. Special thanks go to Mark McKnight who provided technical advice, support, and programming for our Breadboard platform. Support for this research was provided by a grant from the Robert Wood Johnson Foundation, Tata Sons Limited, Tata Consultancy Services Limited, Tata Chemicals Limited, and the National Institute of Social Science. G.I. acknowledges the support by Science Foundation Ireland through grants 17/CDA/4760 and 16/IA/4610.

## Author contributions

H.S., G.I., L.T., and N.C. conceived and designed the research. H.S and G.I. executed the experiment and performed statistical analyses. H.S., G.I., L.T., and N.C. analyzed the results and wrote the paper.

## Additional information

**Competing interests:** The authors declare no competing interests.

