## [Peer Review File · Nature Communications]

Reviewers' comments:

Reviewer #1 (Remarks to the Author):

Authors study sharing in technologically defined social networks. They focus on determinants of human sharing strategy at the collective and individual levels, the impact of network constraints, and on how technical or informational features of the social interactions might foster more equal sharing. Research reveals that the resulting network structure affects the variance in acquisition of the shared resource, and that denser networks decrease any wealth inequality, but this effect diminishes, revealing a saturation property. It is also shown that the benefit of individuals is shaped by their network position, in particular, increasing if they have many partners who in turn have few partners.

How we share knowledge and resources with others, at individual and collective levels, is an intensely investigated subject with obvious practical ramifications. Methods of network science, statistical physics, and social sciences, have been applied successfully and with much effect recently to shed light on the problem from many different perspectives, and also to reveal many different explanations for why we share the way we do. In this sense, the study certainly addresses a relevant setup, and it also delivers results that will surely be of interest to the readership of Nature Communications.

Altogether, I have very much enjoyed reading this manuscript. I find it interesting and clearly written, and satisfying also all the other publication criteria of Nature Communications. The application of network science and game theory is innovative and, as beautifully demonstrated in the manuscript, very useful to facilitate our ability to understand sharing in technologically defined social networks.

The following comments should nevertheless be taken into account.

As far as exploring theoretical models for cooperation equilibrium goes, and the fact that the emergence of cooperation generally depends on how humans respond to their social surroundings, the authors overlook a lot of research done to that effect in other fields. I am not expecting an in-depth overview, but some references ought to be there because as it stands, it seems like not much has been done theoretically since about 20 years. Two useful reviews are Statistical physics of human cooperation, *Physics Reports* 687, 1-51 (2017) and Coevolutionary games - A mini review, *BioSystems* 99, 109-125 (2010) the later in particular related to changing networks and how they affect cooperation.

In terms of original research, the role of information sharing for cooperation has been studied in Information sharing promotes prosocial behaviour, *New J. Phys.* 15, 053010 (2013), as well as in Facilitators on networks reveal optimal interplay between information exchange and reciprocity, *Phys. Rev. E* 89, 042802 (2014). I am sure that these references fit well to the introduction, in particular since the authors build prominently on related game-theoretical research.

In Fig. 2, it is said that wider connection range makes a network dense and decreases economic inequality. Is the first part of this statement not obvious? If the connection range increases then of course the network becomes denser. If I am overlooking something the authors should please clarify this. Otherwise, the fact that denser networks decrease economic inequality has been shown theoretically in numerous research papers dealing with cooperation on scale-free networks with and without assortativity.

Apart from that, I am happy to reiterate my overall positive impressions, and I congratulate the authors to fine research.

Reviewer #2 (Remarks to the Author):

This paper attempts to utilize an AMT-assisted game to guide wireless network resource sharing. Its aim is to introduce network interventions to help more evenly distribute shared resources by controlling network structure. Rich experimental results are presented and discussed, which are impressive. The idea of introducing mTurks (i.e., human) into experiments is interesting in the context of wireless network resource sharing.

Some comments are as follows:

- 1) The concept of sharing economy in wireless network is not new. It appeared more than 7 years ago, e.g., in IEEE Globecom 2011.
- 2) It is not very clear how social networks (as defined in the title) are linked to the physical wireless networks (i.e., WiFi networks in this case). What are the deeper relationships between these two types of networks? How would your experimental design reflect the factors that affect the relationships of these two types of networks? In the real world, social networks (online ones or within the physical society) can use any physical communication networks to carry out their socializing. How to bridge these two types of networks is a challenging and more interesting issue. Sometimes readers get lost as to which type of networks the authors refer to when the paper only says "networks".
- 3) Most importantly, it is not clear what practical impact this paper can make. These WiFi networks under investigation of the paper are typically set up by the home owners without any coordination. Namely there is no centralized controller to adjust the parameters which are tuneable in this paper, e.g., density of the WiFi access points. The paper also mentions that one of its intention is to help more evenly distribute share resources by controlling network structure. However, this is not clear how to make control of network structure exactly. It is impractical to ask a home owner to move their home wireless router from a living room downstairs to a study room upstairs due to various constraints.
- 4) The paper claims that "the individuals' benefit is shaped by their network position". This is the essence of social centrality and it is a known conclusion in the literature of social networks. Probably the authors should have made it clearer how exactly this new discovery differs from the knowns.
- 5) The experimental work is the most salient part of the paper. Rich experimental results have been presented and analysed. However, the method of using AMT to get human involved in the experiments is not novel. And the validity of the experiment results is dependent on not only the game design (which is fine in the paper) but also the subjective choices of the human workers. The game design is more like a simulation so why not simply using a simulation to derive your results, which is simpler and excluding human's subjective and sometimes irresponsible answers?
- 6) It might be a good idea to get researchers of wireless communications/networks background to get involved in this research so as to make it more realistic and useful.

Reviewer #3 (Remarks to the Author):

The Authors design a sharing resources game where each subject can share their unused Internet bandwidth with their neighbours in order to encourage them to share theirs. The aim of the game is to collect as much bandwidth as possible from others. They show that the players behave in reciprocal manner, allocating the bandwidth to the neighbours who allocated their bandwidth to them. The players are placed randomly inside the unit square and they can interact only with the players within their Wifi range. Range had 3 different values in different treatments which leads to 3 different densities of the network. The paper is generally easy to read, although some crucial information is sometimes difficult to find or missing completely.

This is an interesting idea for an experiment and the experiments seems to be properly performed, however the consequent analysis of the results leaves a lot to be desired. Their conclusions cannot

be supported by their analysis. Most of the conclusions can be made directly from the model with random players, without any experiment. They lack a proper theoretical models to show what is the consequence of human behaviour and is just the consequence of the network structure. Furthermore, they do not put their work in a proper perspective, failing to cite many other interesting work on game theory experiments on a network and compare their results with them. Finally, they seem to take the game design too literally. I see this as a good metaphor, however in real world situations this would not be feasible. First I doubt that the people would like to spend time deciding who gets their WIFI and second, the more equal distribution would be easily and more efficiently achieved if the provider would just allocate the unused bandwidth. This is, I believe, already done in the case of, for example, sharing computer resources of the big clusters etc.

Here are more details:

Page 2. "to conducting behavioural experiments involving public goods games 12-15" Only one of the cited papers is not from the authors of this paper and that one is from the year 1986. This creates the false image that the only people doing research in this area are the authors themselves, which is not true and unfair to claim.

Here are some examples of other peoples work:

If they wanted a general reference, this one and the numerous references within would probably be more appropriate:

Chaudhuri, Ananish. "Sustaining cooperation in laboratory public goods experiments: a selective survey of the literature." *Experimental Economics* 14, no. 1 (2011): 47-83.

If they wanted more network oriented experiments:

Gracia-Lázaro, Carlos, Alfredo Ferrer, Gonzalo Ruiz, Alfonso Tarancón, José A. Cuesta, Angel Sánchez, and Yamir Moreno. "Heterogeneous networks do not promote cooperation when humans play a Prisoner's Dilemma." *Proceedings of the National Academy of Sciences* 109, no. 32 (2012): 12922-12926.

This one is on the dynamic network:

Fehl, Katrin, Daniel J. van der Post, and Dirk Semmann. "Co-evolution of behavior and social network structure promotes human cooperation." *Ecology letters* 14, no. 6 (2011): 546-551.

I would especially draw attention of the authors to this paper:

Bendtsen, Kristian Moss, Florian Uekermann, and Jan O. Haerter. "Expert Game experiment predicts emergence of trust in professional communication networks." *Proceedings of the National Academy of Sciences* 113, no. 43 (2016): 12099-12104.

Although the framing of the two papers is different, I find the basic idea very similar. In this experiment the players can provide information to other players, which they cannot use themselves, however if they help other the others will help them. I would suggest to read this paper carefully and compare their results.

A paper from engineering point of view about recourse sharing would be interesting also. The authors should do their research here and point out exactly what is known and not known there and what is their contribution. This paper seems to be solving a more complicated problem in the same area.

Fang, Zuyuan, and Brahim Bensaou. "Fair bandwidth sharing algorithms based on game theory frameworks for wireless ad-hoc networks." In *INFOCOM 2004. Twenty-third Annual Joint Conference of the IEEE Computer and Communications Societies*, vol. 2, pp. 1284-1295. IEEE,

2004.

These are just some examples of the papers. The authors should do their own research of the literature and properly cite the relevant work.

Page 7. Results and Fig. 2

Fig2A - They define the Network Density as a number of ties which was formed divided by the number of ties in the fully connected network, although most of those ties could not be formed, since the players can only interact with the player within the Wifi range. Looking at other results and the snap shots in SI, we see that the most of the links which could have been formed were actually formed. Therefore the "network density" they measure is just the theoretical density of the geometric graph with that radius, which will obviously increase linearly with the radius. Fig 1A shows a trivial result, coming directly from the graph theory which would hold even if the humans would play randomly.

Fig2B similarly just tells us that all the players always allocated all (or maybe most, we cannot tell from this figure) of their resources.

Instead of the Fig 1A would be interesting to see total number of ties which were formed divided by the total number of ties which could have been formed, and instead of Fig 2B, it would be interesting to see the amount of not allocated wealth (wasted wealth)

Fig2C - How does this compare with the random model? Is it possible that Gini coefficient decreases only because there are in average more neighbours in the more dense networks, so the wealth averages out more easily? If we make a simulation in the same networks, with players randomly allocating their bandwidth (just shuffling the contributions of each player), I would expect the same decrease in Gini coefficient with the increase of the density of the network. Simply speaking the sum of 5 random numbers will be less diverse than the sums of 3 random numbers.

Therefore, this result does not really tell us anything about the human behaviour and we never needed an experiment to tell us this. A simple model where people allocate their bandwidth randomly (but not equally to everybody) would probably give the same results. What we want to learn is how reciprocity (which would not be in the model and is observed in the experiment) would change the dynamics of the system. We need to compare the theoretical result to the experiment result to observed the difference.

If we look at the Supplementary Figure S5, it seems that only network visibility condition is influencing human behaviour in this sense. Apparently if people know how many neighbours their neighbours have (and also opportunity to be helped) they act in the way to diminish their these inequality. Interestingly, observing purely how much the neighbour earn, does not change their behaviour. Apparently the player do not make an equality between opportunity (number of neighbours) and wealth.

Fig 3.B -

Network-level reciprocity

What do they exactly call network-level reciprocity?

I am unsure is this effect due to the network or simple direct reciprocity with each individual neighbour. Originally defined network reciprocity (Nowak and May 1992) appears when we have only one action for all of our neighbours and cannot reciprocated individual. Here we can

reciprocate individually, although there are constraints on the whole sum.

Gini

Here the authors do compare their results to the random allocation model, where they assume that players would allocate their resources equally to all their neighbours. We see that in the random model, more dense network has lower Gini coefficient, just as I predicted before. We observe that the Gini coefficient in the experiment is actually lower than that in the random model. This is an interesting result. I would encourage the authors to do more similar analysis. However I would say that the more appropriate null model would be to keep the diversity of the allocation from the experiment, just shuffle the players to which they made the donation and then compare Gini coefficients.

Smaller radius means smaller number of neighbours. So any result which depends on radius actually depends on the number of neighbours. Therefore direct reciprocity is bigger when the radius is bigger. That means that direct reciprocity increases when we have more neighbours. Meaning, the direct reciprocity increases when you have more options.

Would be good to give the definition Graph-theoretic power index in the text, not only in the Figure caption. Or maybe say in the text that the definition is given in the Figure 4 caption.

Fig 4, page 10 in the text

The authors observe that "the geodesic location of a subject largely determines a subject's wealth in a sharing network". This is interesting, however, it seems it would again come just from the network structure, not the human behaviour in it. Would the random model give the same result?

So, their conclusion is that the number of your neighbours' neighbour is important for inequality and we see in SI Fig S5, that when players know the number of their neighbours' neighbours, they Gini coefficient is higher than when they do not know it. It seems that they play in the way to keep the inequality? However the authors claim that both wealth visibility and network visibility do not influence the Gini coefficient. They do not report how did they compare them, but from the graph it looks like the Gini in the network visibility coefficient is different. They should probably make more effort to quantify this difference.

Fig. 4 D Network degree - just looking at the graph it does seem that there is some correlation there, however the authors never report on the p-value. Is this significant? The correlation is bigger for the other two measures, however, the degree alone does seem to have some influence.

Discussion -The first sentence: "We find that manipulable technological changes can increase network density and decrease social inequality." This is a very nice sound bite, however it is also largely inaccurate, overselling of their results.

All their conclusions come from graph theory and could have been made without the experiment. The only behavioural result is that players behave in a reciprocal manner, however this is not new as they themselves say: "cooperative partners generally induce cooperative behavior 13,14,16" (they mostly cite themselves here, however there are other papers claiming the same).

The measure they introduce is interesting, however not a real change from the GPI, which is already previously introduced. The results that "humans seek exploitation" is only reported in the ref 29. where the set up is different enough that we cannot really claim any conflict of the conclusions. However their big conclusion that "humans seek reciprocal exchange" is, as previously

mentions, quite well documented already.

To conclude, the results presented here are not well argued or novel enough for the publication in Nature Communication. However I do find the experiment interesting and I would encourage the authors to re-analyse the results and properly research the literature in order to put their results in the proper context. However that should be a totally new submission.

Reviewer #1 (Remarks to the Author):

Authors study sharing in technologically defined social networks. They focus on determinants of human sharing strategy at the collective and individual levels, the impact of network constraints, and on how technical or informational features of the social interactions might foster more equal sharing. Research reveals that the resulting network structure affects the variance in acquisition of the shared resource, and that denser networks decrease any wealth inequality, but this effect diminishes, revealing a saturation property. It is also shown that the benefit of individuals is shaped by their network position, in particular, increasing if they have many partners who in turn have few partners.

How we share knowledge and resources with others, at individual and collective levels, is an intensely investigated subject with obvious practical ramifications. Methods of network science, statistical physics, and social sciences, have been applied successfully and with much effect recently to shed light on the problem from many different perspectives, and also to reveal many different explanations for why we share the way we do. In this sense, the study certainly addresses a relevant setup, and it also delivers results that will surely be of interest to the readership of Nature Communications.

Altogether, I have very much enjoyed reading this manuscript. I find it interesting and clearly written, and satisfying also all the other publication criteria of Nature Communications. The application of network science and game theory is innovative and, as beautifully demonstrated in the manuscript, very useful to facilitate our ability to understand sharing in technologically defined social networks.

-- We are grateful for these positive comments, and for the detailed and helpful suggestions below. We have endeavored to address all of them in the manuscript and SI.

The following comments should nevertheless be taken into account.

As far as exploring theoretical models for cooperation equilibrium goes, and the fact that the emergence of cooperation generally depends on how humans respond to their social surroundings, the authors overlook a lot of research done to that effect in other fields. I am not expecting an in-depth overview, but some references ought to be there because as it stands, it seems like not much has been done theoretically since about 20 years. Two useful reviews are Statistical physics of human cooperation, *Physics Reports* 687, 1-51 (2017) and Coevolutionary games - A mini review, *BioSystems* 99, 109-125 (2010) the later in particular related to changing networks and how they affect cooperation.

In terms of original research, the role of information sharing for cooperation has been studied in Information sharing promotes prosocial behaviour, *New J. Phys.* 15, 053010 (2013), as well as in Facilitators on networks reveal optimal interplay between information exchange and reciprocity, *Phys. Rev. E* 89, 042802 (2014). I am sure that these references fit well to the introduction, in particular since the authors build prominently on related game-theoretical research.

-- We thank R1 for suggesting this other valuable work, which we now cite as Ref. [15]. [20], and [55]. We are familiar, too, with the work on cooperation on graphs, and have contributed to that literature ourselves over the past decade.

In Fig. 2, it is said that wider connection range makes a network dense and decreases economic inequality. Is the first part of this statement not obvious? If the connection range increases then of course the network becomes denser. If I am overlooking something the authors should please clarify this. Otherwise, the fact that denser networks decrease economic inequality has been shown theoretically in numerous research papers dealing with cooperation on scale-free networks with and without assortativity.

-- We agree with the point that the density result is nothing special. We moved the fact about network density to the section regarding experimental settings. We also changed Figure 2 so as to show more clearly the diminishing improvement in sharing inequality with increases in network density. We added statistical analysis to confirm the nonlinearity (Supplementary Table S1). As R1 pointed out, here we *empirically confirmed* the theoretical prediction that denser networks decrease economic inequality. We also showed the additional finding that the change in economic inequality is diminishing, which is important in practical applications.

Apart from that, I am happy to reiterate my overall positive impressions, and I congratulate the authors to fine research.

-- We thank R1 for the generally favorable comments. We have tried to respond comprehensively to all the comments and to improve the paper accordingly.

Reviewer #2 (Remarks to the Author):

This paper attempts to utilize an AMT-assisted game to guide wireless network resource sharing. Its aim is to introduce network interventions to help more evenly distribute shared resources by controlling network structure. Rich experimental results are presented and discussed, which are impressive. The idea of introducing mTurks (i.e., human) into experiments is interesting in the context of wireless network resource sharing.

-- We are grateful for these positive comments, and for the detailed and helpful suggestions below. We have endeavored to add all of them in the manuscript and SI. And we have also worked to clarify this study's relevance.

Some comments are as follows:

1) The concept of sharing economy in wireless network is not new. It appeared more than 7 years ago, e.g., in IEEE Globecom 2011.

-- We agree that resource sharing in wireless networks has been an active research subject for many years. For instance in the IEEE Globecom 2011 conference, as R2 suggests, there are several papers about spectrum sharing, or cooperative P2P architectures and so on. And recently, this idea has gained new momentum, with suggestions for mobile Internet sharing, Wi-Fi sharing communities, energy sharing in smart grid, etc.. Indeed, there are several studies that propose optimization algorithms that maximize the sharing benefits, or game-theoretic frameworks that model the users' decisions and predict the equilibriums of their interactions. We have contributed to this literature, as this is one of our main research activities (for example: <http://globecom2015.ieee-globecom.org/content/tutorials#TT-13>).

However all these papers (including our previous papers) use stylized theoretical non-cooperative or cooperative game-theoretic models for capturing the user behavior, assuming fully rational users. Therefore, they do not consider (nor provide evidence) about how users actually make sharing decisions, how the underlying network structure affects their interactions, and how information availability impacts their behavior. On the other hand, several recent studies in the social sciences have showed experimentally that such biases and effects are prevalent in human decision-making (e.g. Ref [35]); and so here we make the natural next step of investigating similar effects in sharing economy networks. We therefore believe that our work fills an important gap in engineering.

Following the R2's comment, we now cite and discuss additional engineering studies for resource sharing in wireless (and other) networks (Ref. [26], [27], [28], [29], [48], and [49]), and explain in detail our contribution in relation to these prior studies.

2) It is not very clear how social networks (as defined in the title) are linked to the physical wireless networks (i.e., WiFi networks in this case). What are the deeper relationships between these two types of networks? How would your experimental design reflect the factors that affect the relationships of these two types of networks? In the real world, social networks (online ones or within the physical society) can use any physical communication networks to carry out their socializing. How to bridge these two types of networks is a challenging and more interesting issue.

Sometimes readers get lost as to which type of networks the authors refer to when the paper only says “networks”.

-- We use these different terms in order to distinguish between the underlying wireless network that creates the possibilities for social interactions (here, exchange of resources), and the social network to describe the actual realized interactions. In other words, the wireless network here provides opportunities for cooperation among the users that the users (being truly independent decision makers in our game) can leverage or not in order to develop a social tie. Please note that here the resource exchanges are indeed social interactions, and this is manifested by the fact that they convey social value for the participants (an important finding of our work).

As R2 correctly suggests, these two layers of networks “interact,” and it is important (and interesting) to reveal the underlying mechanism. To this end, we leverage different experimental treatments where we change the physical network structure (by changing the node radius) or (distinctly) the information the players have at their disposal. We observe that, indeed, these changes in the wireless network affect the actions of the users, resulting in different types of social interactions (or, social network structures). We understand that the presentation of this idea can be improved and we have tried in the revised paper to add clarity in this discussion.

We have updated the discussion to clarify and highlight the differences and relations among the two types of networks. We also have included pointers to figures in the SI which show that the social interactions create a network graph that it is often different than the wireless network graph (Supplementary Figure S2).

3) Most importantly, it is not clear what practical impact this paper can make. These WiFi networks under investigation of the paper are typically set up by the home owners without any coordination. Namely there is no centralized controller to adjust the parameters which are tuneable in this paper, e.g., density of the WiFi access points. The paper also mentions that one of its intention is to help more evenly distribute share resources by controlling network structure. However, this is not clear how to make control of network structure exactly. It is impractical to ask a home owner to move their home wireless router from a living room downstairs to a study room upstairs due to various constraints.

-- As we described in the discussion section, one possible implication is to restrict the sharing options of users. Our study shows that not only network density, but also degree assortativity, affects sharing inequality among participants (Table 1). Thus, sharing centrality can be improved by *cutting* some links to improve degree assortativity even within certain access ranges. This is a novel and practical idea for reducing sharing inequality. We think that a WiFi service company can control the options that users see in sharing because they are likely to share their WiFi through the company’s web or mobile app. We also showed that neighbors’ information disclosure might not affect the overall macro situation (Supplementary Figure S9 and S10). As such information control is one of typical practical implications in sharing services, we believe the result with information visibility also makes practical impact. Finally, our work suggests ways companies might strategically add nodes to the graph.

We believe our work brings in light important experimental results about user behavior in WiFi networks and other sharing economy applications and that this might have a multifaceted practical impact. We have updated the discussion to clarify the design principles and practical guidelines that stem from this work, both for the specific WiFi network sharing example, but also for other sharing network instances.

4) The paper claims that “the individuals’ benefit is shaped by their network position”. This is the essence of social centrality and it is a known conclusion in the literature of social networks. Probably the authors should have made it clearer how exactly this new discovery differs from the knowns.

-- As R2 points out, “individual’s benefit is shaped by their network position” is the essence of social centrality. However, what structural property represents individual’s benefit has been unclear in sharing networks *per se*. Thus, we tested several centrality measures with individual wealth, and also introduced a new, simple centrality measure (Figure 5), which we call “sharing centrality.” For example, if we had evaluated only the standard centrality measures, such as degree centrality and betweenness centrality, we would have got a wrong conclusion that individual’s benefit is *not* shaped by their network position in sharing networks because they are little correlated with individual wealth. We think that Figure 5 shows how sharing-network-specific centralities differ from the known ones. In response to the R2’s comment, we have clarified the advantages of “sharing centrality” *per se*.

5) The experimental work is the most salient part of the paper. Rich experimental results have been presented and analysed. However, the method of using AMT to get human involved in the experiments is not novel. And the validity of the experiment results is dependent on not only the game design (which is fine in the paper) but also the subjective choices of the human workers. The game design is more like a simulation so why not simply using a simulation to derive your results, which is simpler and excluding human’s subjective and sometimes irresponsible answers?

-- We believe that using experiments and studying actual human behavior is instrumental in revealing actual effects, verifying (or not) our theoretical assumptions or model-building exercises. They are also useful for developing novel experimentally-validated models and theories. Online experiments offer the unique opportunity to receive actual feedback from thousands of participants. This technique also can isolates players in this game setting; hence, we are able to observe their strategies without being biased from other factors not related to the test.

However, we agree with R2 that simulations play a crucial role as well, and we have included several such studies (Figure 3, 4 and Supplementary Figure S8) in our study. These complement our experimental work and are also used to compare our findings with prior related theoretical models (in engineering and social science).

In fact, by comparing actual human behaviors and theoretical expectations with simulations, we find that no simulation results firmly predicted our experiment outcomes.

If we excluded human's subjective actions, we would not expect the dynamics of wealth inequality and reciprocity (Figures 3 and 4). If we expected only reciprocity in human behaviors, we would not see any structural impacts on individual wealth (Supplementary Figure S8). We believe that, without the experiments within this simplified model, we could not obtain or noteworthy findings in network exchange and resource sharing.

6) It might be a good idea to get researchers of wireless communications/networks background to get involved in this research so as to make it more realistic and useful.

-- We agree with R2 that this inherently interdisciplinary study can only be successful if engineers and sociologists collaborate, which is exactly the case here: the author list includes 2 sociologists, and 2 wireless network researchers. The latter have conducted several theoretical and experimental studies for wireless networks. L. Tassiulas (IEEE Fellow, IEEE INFOCOM Achievement Award 2007, IEEE Kobayashi Award 2016, Chair of Electrical Engineering, etc.) has made seminal contributions in cooperative wireless communications which have opened new research areas and are applied to industry. Also, G. Iosifidis has been working on cooperative networks and network economics for more than 10 years, proposing game-theoretical models and implementing prototypes. In fact, the idea of this work has emerged after our long experience in studying, as engineers, such systems; and also discussing their limitations with several industry partners (in various EU and US projects). On the other hand, N. Christakis and H. Shirado are spearheading the study of cooperation in social networks, using novel datasets and pioneering experimental methods. To the best of our knowledge, ours is a rare interdisciplinary collaboration that brings together expertise on cooperative systems and networks from engineering and social sciences.

Reviewer #3 (Remarks to the Author):

The Authors design a sharing resources game where each subject can share their unused Internet bandwidth with their neighbours in order to encourage them to share theirs. The aim of the game is to collect as much bandwidth as possible from others. They show that the players behave in reciprocal manner, allocating the bandwidth to the neighbours who allocated their bandwidth to them. The players are placed randomly inside the unit square and they can interact only with the players within their Wifi range. Range had 3 different values in different treatments which leads to 3 different densities of the network. The paper is generally easy to read, although some crucial information is sometimes difficult to find or missing completely.

This is an interesting idea for an experiment and the experiments seems to be properly performed, however the consequent analysis of the results leaves a lot to be desired. Their conclusions cannot be supported by their analysis. Most of the conclusions can be made directly from the model with random players, without any experiment. They lack a proper theoretical models to show what is the consequence of human behaviour and is just the consequence of the network structure. Furthermore, they do not put their work in a proper perspective, failing to cite many other interesting work on game theory experiments on a network and compare their results with them. Finally, they seem to take the game design too literally. I see this as a good metaphor, however in real world situations this would not be feasible. First I doubt that the people would like to spend time deciding who gets their WIFI and second, the more equal distribution would be easily and more efficiently achieved if the provider would just allocate the unused bandwidth. This is, I believe, already done in the case of, for example, sharing computer resources of the big clusters etc.

-- We are grateful for the detailed and helpful suggestions below. We have endeavored to address all of them, through new simulations, new analyses, and new references in the manuscript and SI. In particular, we've worked to clarify on the difference between structural and behavioral effects.

We would like to emphasize that we use simulations and null models to set proper benchmarks against which we compare our findings (and having included new simulations following R3 suggestions); our work is based on previous theoretical models that we have analyzed mathematically (we provide more details in this revised version); and that the treatments we employ serve as interventions, and our goal is exactly what R3 suggests: to analyze the actual human behavior and investigate the impact of the network properties.

We selected this experiment as it has the main features of actual business applications. For instance, the very popular FON wireless community uses WiFi routers with increased range to help users share their Internet connections. Telecommunication providers (e.g., British Telecom, Deutsche Telekom, etc.) use it to complement their Internet services. Other companies, like Comcast, have developed their own WiFi sharing services (e.g., xfinity) in order to increase the users access to Internet but also reduce their networks' congestion. It is important to stress that users are typically not willing to accept externally imposed solutions. For example, there were several complaints (and legal suits) from xfinity customers against Comcast for making available their home WiFi routers to others, without their approval. Finally, we note that an implementation of the WiFi sharing scheme we consider here would not actually require the users to make sharing decisions every day. The granularity can be coarser (e.g.,

every week) or even can be automated through a proper software proxy that implements the users' sharing preferences.

In view of all these actual situations, we believe that the obtained results can be possibly extended to other similar systems. Moreover, the specific problem is actually of high interest to wireless network engineers and the industry, and there are already practical applications quite close to the hypothetical scenario we have used in AMT. Besides, in our prior work, we have designed and implemented practical working engineering prototypes for even more sophisticated wireless connectivity sharing scenarios (see Ref. [35]).

Here are more details:

Page 2. "to conducting behavioural experiments involving public goods games 12-15" Only one of the cited papers is not from the authors of this paper and that one is from the year 1986. This creates the false image that the only people doing research in this area are the authors themselves, which is not true and unfair to claim.

Here are some examples of other peoples work:

If they wanted a general reference, this one and the numerous references within would probably be more appropriate:

Chaudhuri, Ananish. "Sustaining cooperation in laboratory public goods experiments: a selective survey of the literature." *Experimental Economics* 14, no. 1 (2011): 47-83.

If they wanted more network oriented experiments:

Gracia-Lázaro, Carlos, Alfredo Ferrer, Gonzalo Ruiz, Alfonso Tarancón, José A. Cuesta, Angel Sánchez, and Yamir Moreno. "Heterogeneous networks do not promote cooperation when humans play a Prisoner's Dilemma." *Proceedings of the National Academy of Sciences* 109, no. 32 (2012): 12922-12926.

This one is on the dynamic network:

Fehl, Katrin, Daniel J. van der Post, and Dirk Semmann. "Co-evolution of behavior and social network structure promotes human cooperation." *Ecology letters* 14, no. 6 (2011): 546-551.

I would especially draw attention of the authors to this paper:

Bendtsen, Kristian Moss, Florian Uekermann, and Jan O. Haerter. "Expert Game experiment predicts emergence of trust in professional communication networks." *Proceedings of the National Academy of Sciences* 113, no. 43 (2016): 12099-12104.

Although the framing of the two papers is different, I find the basic idea very similar. In this experiment the players can provide information to other players, which they cannot use themselves, however if they help other the others will help them. I would suggest to read this paper carefully and compare their results.

-- We thank R3 for suggesting these valuable papers, which we now cite as Ref. [16], [17], [21], and [39]. We have also added relatively recent reviews (Ref. [14] and [20]). As

R3 points out, there are rich theoretical and empirical backgrounds in networked human cooperation.

However, we would like to note that our research problem (i.e., network exchange) is fundamentally different from the typical social dilemma's problem (e.g., Ref. [33] has clarified the difference). Also, please note that the Expert Game, Ref [39], studies an information exchange problem which, however, does not have any type of resource constraints. Resource constraints are a basic structural ingredient of sharing economy experiments and are central to our study (see Supplementary Figure S1).

A paper from engineering point of view about resource sharing would be interesting also. The authors should do their research here and point out exactly what is known and not known there and what is their contribution. This paper seems to be solving a more complicated problem in the same area.

Fang, Zuyuan, and Brahim Bensaou. "Fair bandwidth sharing algorithms based on game theory frameworks for wireless ad-hoc networks." In INFOCOM 2004. Twenty-third Annual Joint Conference of the IEEE Computer and Communications Societies, vol. 2, pp. 1284-1295. IEEE, 2004.

These are just some examples of the papers. The authors should do their own research of the literature and properly cite the relevant work.

-- Thank you for this important suggestion. We regret that we haven't included a more detailed discussion about engineering works in this paper, and we have corrected this in our revised paper including R3's suggested paper, as Ref. [26], [27], [28], [29], [48], and [49]. We are quite familiar with this literature.

One very important point missing in all these studies (and all related engineering studies actually) is the assumption of a user who makes rational utility-maximizing sharing decisions. The first goal of this paper was to investigate exactly this assumption, and create the basis for more realistic engineering studies in the future. Furthermore, there are no prior works studying the impact of the network structure on the resource inequality, how information availability affects human decisions, and how the node position (centrality) impacts the subjects earnings in such sharing networks. These are all contributions of our work.

Page 7. Results and Figure 2

Fig2A - They define the Network Density as a number of ties which was formed divided by the number of ties in the fully connected network, although most of those ties could not be formed, since the players can only interact with the player within the Wifi range. Looking at other results and the snap shots in SI, we see that the most of the links which could have been formed were actually formed. Therefore the "network density" they measure is just the theoretical density of the geometric graph with that radius, which will obviously increase linearly with the radius. Fig 1A shows a trivial result, coming directly from the graph theory which would hold even if the humans would play randomly.

Fig2B similarly just tells us that all the players always allocated all (or maybe most, we cannot

tell from this figure) of their resources.

Instead of the Fig 1A would be interesting to see total number of ties which were formed divided by the total number of ties which could have been formed, and instead of Fig 2B, it would be interesting to see the amount of not allocated wealth (wasted wealth)

-- We agree with the point that the density and wealth result is nothing special. We moved the fact about network density to the section regarding experimental settings. We also moved the result of wealth to the SI and added that of wasted wealth, which R3 suggests (Supplementary Figure S3). Please note that we use the standard definition of network density that is quantified as the ratio of existing ties over the number possible ties among all pairs of nodes in a network.

We also show the total symmetric shared resources divided by the total shared resources per link as “network-level reciprocity” in Figure 4. As this study’s networks are weighted with continuous values (i.e., the amount of shared resources), it is less appropriate to evaluate their tie activation as binary values, such as whether or not a tie is formed by an interaction. We believe that reciprocity shown in Figure4 can address (in part) what R3 suggests with respect to measuring network density.

Fig2C - How does this compare with the random model? Is it possible that Gini coefficient decreases only because there are in average more neighbours in the more dense networks, so the wealth averages out more easily? If we make a simulation in the same networks, with players randomly allocating their bandwidth (just shuffling the contributions of each player), I would expect the same decrease in Gini coefficient with the increase of the density of the network. Simply speaking the sum of 5 random numbers will be less diverse than the sums of 3 random numbers.

Therefore, this result does not really tell us anything about the human behaviour and we never needed an experiment to tell us this. A simple model where people allocate their bandwidth randomly (but not equally to everybody) would probably give the same results. What we want to learn is how reciprocity (which would not be in the model and is observed in the experiment) would change the dynamics of the system. We need to compare the theoretical result to the experiment result to observed the difference.

If we look at the Supplementary Figure S5, it seems that only network visibility condition is influencing human behaviour in this sense. Apparently if people know how many neighbours their neighbours have (and also opportunity to be helped) they act in the way to diminish their these inequality. Interestingly, observing purely how much the neighbour earn, does not change their behaviour. Apparently the player do not make an equality between opportunity (number of neighbours) and wealth.

-- We fully agree with R3 that, in order to assess an empirical effect, it is necessary to employ proper null models and theoretical predictions. In response to R3’s suggestion, we show the comparison of actual Gini coefficients with that of null model with the random-shuffle sequence, which R3 suggests, in Figure 3. We also clarify each impact of human behaviors and network structure on sharing outcomes in the manuscript.

Figure 3 shows that, as the radius increases, the Gini reduces for both the actual experiment and the null model (random reshuffling). However, this improvement is not linear and actually diminishes fast (compare 2nd and 3rd panels), unlike the hypothesis suggested by R3. Also, we see that, in all three cases, the actual game has much lower Gini, compared to the null model. We think that our result clearly presents the behavioral effects, which we couldn't know without experiments. We are very grateful for this suggestion.

Fig 3.B -

Network-level reciprocity

What do they exactly call network-level reciprocity?

I am unsure is this effect due to the network or simple direct reciprocity with each individual neighbour. Originally defined network reciprocity (Nowak and May 1992) appears when we have only one action for all of our neighbours and cannot reciprocated individual. Here we can reciprocate individually, although there are constrains on the whole sum.

-- As we have described above, the networks we study here are weighted, and the actors' decisions are modeled with continuous values (i.e., the amount of shared resources). Hence, the originally network reciprocity (which the Reviewer correctly points to) cannot be applied here directly. Instead, we calculated the network-level reciprocity using a method proposed in Ref. [41]. We added the measurement details to the caption of Figure 4. We also have confirmed that other measurements give us very similar results.

Gini

Here the authors do compare their results to the random allocation model, where they assume than players would allocate their resources equally to all their neighbours. We see that in the random model, more dense network has lower Gini coefficient, just as I predicted before. We observe that the Gini coefficient in the experiment is actually lower that in the random model. This is an interesting results. I would encourage the authors to do more similar analysis. However I would say that the more appropriate null model would be to keep the diversity of the allocation from the experiment, just shuffle the players to which they made the donated and then compare Gini coefficients.

-- We thank R3 for suggesting another null model to evaluate the structural effect separately from the behavioral one. We did the simulation with the random-shuffle sequence and compared the hypothetical Gini coefficients with the actual ones in the new Figure 3. The results are very similar to those of our original null model in that subjects equally allocate their resources; we show the results of both null models in Supplementary Figure S4. In response to R3's suggestion, we clarified the separate impact of network structure and human behaviors with an additional paragraph and a new Figure 3. Also, we contacted additional analysis to evaluate the correlation between the final Gini coefficients of the experiment and the simulation (the correlation is 0.872), described in the manuscript. The comparison with the null models shows that the actual human subjects significantly reduce the wealth inequality over their sharing transactions

(Figure 3). Also, network structure is associated with the difficulty in inequality improvement.

Smaller radius mean smaller number of neighbours. So any result which depend on radius actually depends on the number of neighbours. Therefore Direct reciprocity is bigger when the radius is bigger. That means that direct reciprocity increases when we have more neighbours. Meaning, the direct reciprocity increases when you have more options.

-- We partially agree with this point. However, the experiment also shows that the improvement due to the connection radius (i.e., the number of neighbor or network density) is diminishing with increase in it (Figure 4B). To clarify the diminishing impact at the individual level, we added the additional result as Supplementary Figure S5. The result shows that the direct reciprocity does not always increase when subjects have more options.

Would be good to give the definition Graph-theoretic power index in the text, not only in the Figure caption. Or maybe say in the text that the definition is given in the Figure 4 caption.

-- Thank you for the suggestion. We had a detailed explanation in the Figure 5 (which is the original Figure 4) caption and mentioned this in the body of part.

Fig 4, page 10 in the text

The authors observe that "the geodesic location of a subject largely determines a subject's wealth in a sharing network". This is interesting, however, it seem it would again come just from the network structure, not the human behaviour in it. Would the random model give the same result?

-- All the centralities, including standard ones such as degree centrality or betweenness centrality, are calculated based solely on the network structure, and not based on the actors' strategy. That is, by definition, they do not consider/account for the human behavior. Besides, the purpose of centrality measures is to identify (and quantify) a key structural property that captures the individual's advantage in certain social interactions.

In this study we find that the GPI and sharing centrality are *more* appropriate structural measurements than other standard centralities in sharing networks (Figure 5). Note that GPI has not been initially designed for this sharing economy game, and hence this is a non-trivial finding. Even more so, we find that a simpler-to-calculate metric (sharing centrality) has similar (or even higher) predictive power.

We thank R3 for suggesting the comparison with a random model in the centrality measurements. We found that sharing centrality represents the structural effect even more clearly than GPI. We added a description of the benefits of sharing centrality to the manuscript and Supplementary Figure S8.

So, their conclusion is that the number of your neighbours' neighbour is important for inequality and we see in SI Fig S5, that when players know the number of their neighbours' neighbours, they Gini coefficient is higher than when they do not know it. It seems that they play in the way to keep the inequality? However the authors claim that both wealth visibility and network visibility do not influence the Gini coefficient. They do not report how did they compare them, but from the graph it looks like the Gini in the network visibility coefficient is different. They should probably make more effort to quantify this difference.

-- We confirmed with statistical analysis (incorporating the additional information as dummy variables) that there is no statistically significant impact of additional information on Gini coefficient when controlling for the impact of network structure. We added Supplementary Table S1 to show the statistical result and removed the original Figure S5 for the avoidance of confusion.

Fig. 4 D Network degree - just looking at the graph is does seems that there is some correlation there, however the authors newer report on the p-value. Is this significant? The correlation is bigger for the other two measure, however, the degree alone does seem to have some influence.

-- We added the p-value results in the caption of Figure 5 (originally Figure 4), which show all the coefficients are statistically significant ($P < 0.05$). Given the significance, we toned down the sentence about the correlation in the manuscript.

Also, R3's insight that the degree also does have some influence is right. In fact, sharing centrality incorporates the influence of ego's degree in its calculation. That is, if the neighbors of an individual have the same number of neighbors, the individual's sharing centrality is identical to her degree. From this perspective, sharing centrality includes the negative impact of neighbor's degree in addition to the positive impact of self degree, which improves the prediction of individual wealth.

Discussion -The first sentence: "We find that manipulable technological changes can increase network density and decrease social inequality." This is a very nice sound bite, however it is also largely inaccurate, overselling of their results.

-- We toned down the sentence.

All their conclusions come from graph theory and could have been made without the experiment. The only behavioural result is that players behaviour in reciprocal manner, however this is not new as they themselves say: "cooperative partners generally induce cooperative behavior 13,14,16" (they mostly cite themselves here, however there are other papers claiming the same).

The measure they introduce is interesting, however not a real change from the GPI, which is already previously introduced. The results that "human seek exploitation" is only reported in the ref 29. where the set up is different enough that we cannot really claim any conflict of the conclusions. However their big conclusion that "human seek reciprocal exchange" is, as previously mentions, quite well documented already.

-- In response to R3's suggestion, we clarified the (different) impact of network

structure and of human behavior on the resource sharing outcome, updating accordingly the text in the manuscript and also the figures.

We believe that the new proposed metric of sharing centrality has several advantages compared to GPI. Namely, its calculation requires lightweight computations, using only local information about sharing decisions of nodes, and this is particularly important for the large and dynamic networks that most often arise in sharing economy applications. In fact, this simple metric helped us to approach the solution of sharing inequality (Table 1); we cannot easily imagine how the complex GPI could have directed us to such a solution. Also, this sharing centrality metric captures more accurately the network structure's impact on individual wealth separately from behavioral changes (Supplementary Figure S8).

We also removed the part of theoretical comparison between exploitation and reciprocation. Instead, we show details of actual human behaviors, for example the reciprocity level across network degree (Supplementary Figure S5). With the experiments, we are able to confirm people's reciprocity in sharing networks with structural and resource constraints (that is a key character of resource sharing). We also find that, even though humans reduce the wealth gap between them with their reciprocal disposition, network structure also affects their wealth (Figure 5 and Supplementary Figure S8) due to the competition between alters caused by resource constraint (Supplementary Figure 1). We believe that we would not share their firm, significant knowledge in network exchange and resource sharing without this actual experiment.

To conclude, the results presented here are not well argued or novel enough for the publication in Nature Communication. However I do find the experiment interesting and I would encourage the authors to re-analyse the results and properly research the literature in order to put their results in the proper context. However that should be a totally new submission.

-- We thank R3 for the careful review of our paper and for the constructive feedback. We have worked to address each comment carefully by re-analyzing the results, comparing with prior theoretical and empirical studies, implementing novel simulations, and updating the references.

Reviewers' comments:

Reviewer #1 (Remarks to the Author):

The authors have revised their manuscript comprehensively and with love to detail. I warmly recommend publication in Nature Communications in present form.

Reviewer #2 (Remarks to the Author):

Many thanks for the effort made by the authors in the revision. Most of my concerns in the previous round of review have been addressed satisfactorily. However the following issues remain:

1) Re: my previous comment #2. "It is not very clear how social networks (as defined in the title) are linked to the physical wireless networks (i.e., WiFi networks in this case). What are the deeper relationships between these two types of networks? How would your experimental design reflect the factors that affect the relationships of these two types of networks? In the real world, social networks (online ones or within the physical society) can use any physical communication networks to carry out their socializing. How to bridge these two types of networks is a challenging and more interesting issue."

Though the revision has provided a graph (S2) for this purpose the graph itself still does not have too much bearing of the physical wireless network itself. The graph remains focused on social features of networks. In social networks a user in US might well talk to a friend in the UK, which is only one hop as far as social connection is concerned. However, there are many hops to connect the UK and US in the physical communication networks. How exactly can this disparity can be described or even modelled is still not addressed by the revised manuscript.

2) Re: my previous comment #3. "3) Most importantly, it is not clear what practical impact this paper can make. These WiFi networks under investigation of the paper are typically set up by the home owners without any coordination. Namely there is no centralized controller to adjust the parameters which are tuneable in this paper, e.g., density of the WiFi access points. The paper also mentions that one of its intention is to help more evenly distribute share resources by controlling network structure. However, this is not clear how to make control of network structure exactly. It is impractical to ask a home owner to move their home wireless router from a living room downstairs to a study room upstairs due to various constraints."

I am glad that the revision has discussed about WiFi service providers. However, there are still two cases missing: 1) WiFi access points/routers from different service providers; 2) the wifi APs set up by households themselves rather than service providers. Note that WiFi operates on unlicensed spectrum band, which means anybody can stick up an AP at any place they like. These APs are totally out of the control of any service providers.

Reviewer #3 (Remarks to the Author):

The author invested a serious effort to answer my comments. They made a null model and compare their results with it, showing what is the impact of human behavior and what comes out directly from the graph theory. Furthermore, they put their work in proper perspective, citing the proper literature and showing what is actually innovative about this paper.

I am in general satisfied with the changes and I do think that paper is now scientifically sound.

A couple of minor comments:

- This sentence seems to need updating. Figure 2 is not what this sentence says, and it does not go with other changes: "the network density (the fraction of ties present in the network versus the number of all possible ties) increases monotonically with the connection range (Figure 2)."

- How come the experiments have the same lengths for different radii, but different lengths for different setups?

- Supplementary Figure S3:

What is the definition of the wasted resources. Total amount which is not allocated? Somebody actually does not allocate all of their resources? Did you check who are the people who waste their resources? Could it be that those are the people who have only one neighbor and feel like that neighbor is not giving them enough? In that case it make sense that with the increase of the network the resources are less wasted, since it is lees likely that nodes have only one neighbor.

Talking about that, did you analyze more deeply the behavior of the people who can have only one neighbor and who have more neighbors.

- Supplementary Figure S4: The legend would make understanding of the figure much easier.

Typo: Page in SI "realizesd"

Reviewers' comments:

Reviewer #1 (Remarks to the Author):

The authors have revised their manuscript comprehensively and with love to detail. I warmly recommend publication in Nature Communications in present form.

-- We thank R1 for the favorable comments and recommendation.

Reviewer #2 (Remarks to the Author):

Many thanks for the effort made by the authors in the revision. Most of my concerns in the previous round of review have been addressed satisfactorily. However the following issues remain:

1) Re: my previous comment #2. “It is not very clear how social networks (as defined in the title) are linked to the physical wireless networks (i.e., WiFi networks in this case). What are the deeper relationships between these two types of networks? How would your experimental design reflect the factors that affect the relationships of these two types of networks? In the real world, social networks (online ones or within the physical society) can use any physical communication networks to carry out their socializing. How to bridge these two types of networks is a challenging and more interesting issue.” Though the revision has provided a graph (S2) for this purpose the graph itself still does not have too much bearing of the physical wireless network itself. The graph remains focused on social features of networks. In social networks a user in US might well talk to a friend in the UK, which is only one hop as far as social connection is concerned. However, there are many hops to connect the UK and US in the physical communication networks. How exactly can this disparity can be described or even modelled is still not addressed by the revised manuscript.

-- We understand this point. In this study, we regarded “technical networks” as the actual wireless links between the Wi-Fi routers that are technically possible (or, to be more precise, between the different users and the Wi-Fi routers), and “social networks” as the actual social interactions with sharable bandwidth that owners of the routers agree to serve each other’ needs, or the ties that are thus realized.

Our experiments reveal that the technological network shapes (to large extent) the social interactions. In other words, whether two users who are in range (connected in the technological network) will actually exchange resources depends on their degree, sharing centrality, preferences, and so on. To make this distinction between the potential (technically defined) and the actual (socially

realized) networks more clear, we updated Supplementary Figure S2. We also clarified the difference in the main text.

In our game scenario, subjects' sharable resources (Wi-Fi access; i.e., the capital of social exchange) are physically constrained. Their social activities – which they can choose freely – are limited to their physically-close neighbors. However, we agree with R2 that some overlays between technological and social layers are not physically constrained (e.g., p2p overlays, virtual private networks, virtual network embedding, etc.). In those cases, a link in the social layer graph (the social network) does not have necessarily to correspond to a link in the lower layer technological network. Although we used physically-constrained networks as an example, the logic and results of sharing centrality are not limited to the physically-constrained sharing networks. The sharing centrality can be a useful indicator in physically-unconstrained sharing networks too.

2) Re: my previous comment #3. “3) Most importantly, it is not clear what practical impact this paper can make. These WiFi networks under investigation of the paper are typically set up by the home owners without any coordination. Namely there is no centralized controller to adjust the parameters which are tuneable in this paper, e.g., density of the WiFi access points.

-- Thank you for this comment that gives us the opportunity to further clarify this important issue. We agree with R2 that, in these sharing scenarios, there is no centralized controller. In some ways, this is another point of our set-up. This makes it imperative to study how humans, who are free to manage their own resources, make their sharing decisions, and how the underlying network structure or the information availability affects their strategies.

Now, the devices' capabilities are typically determined by the technology. For instance, the range of a Wi-Fi router depends on the protocol it employs or the antenna gain, e.g., FON routers have improved antennas in order to exactly facilitate sharing. Of course the users can remove the antennas, switch off their routers, and so on. These non-cooperative actions are actually captured by our study by the “resource-waste” decision.

The paper also mentions that one of its intention is to help more evenly distribute share resources by controlling network structure. However, this is not clear how to make control of network structure exactly. It is impractical to ask a home owner to move their home wireless router from a living room downstairs to a study room upstairs due to various constraints.”

I am glad that the revision has discussed about WiFi service providers. However, there are still two cases missing: 1) WiFi access points/routers from different service providers;

2) the wifi APs set up by households themselves rather than service providers. Note that WiFi operates on unlicensed spectrum band, which means anybody can stick up an AP at any place they like. These APs are totally out of the control of any service providers.

-- Our scenario does not consider device incompatibility directly, yet this can be also captured by the absence of links. In other words, two incompatible routers will not be able to connect and this will restrict the sharing/exchange of resources. Our objective was not to emulate *every* feature of technologically defined networks, but rather to develop a realistic, usable game (that captured many pertinent features) in order to explore social behavior. In addition, Wi-Fi service providers cannot control the Wi-Fi access points as R2 pointed out, but they could control whether their Wi-Fi resources are sharable or not. For instance, we assume that Wi-Fi providers can intervene in sharing networks by manipulating sharing partners displayed in the application of service providers.

We have added these implementation examples, including the devices' capabilities, to the main text.

Reviewer #3 (Remarks to the Author):

The author invested a serious effort to answer my comments. They made a null model and compare their results with it, showing what is the impact of human behavior and what comes out directly from the graph theory. Furthermore, they put their work in proper perspective, citing the proper literature and showing what is actually innovative about this paper.

I am in general satisfied with the changes and I do think that paper is now scientifically sound.

-- We are grateful for the favorable comments and careful re-reviews below. Given R3's suggestions, we revised our explanation to make it clear.

A couple of minor comments:

- This sentence seems to need updating. Figure 2 is not what this sentence says, and it does not go with other changes: "the network density (the fraction of ties present in the network versus the number of all possible ties) increases monotonically with the connection range (Figure 2)."

-- We removed the reference of Figure 2 in the sentence.

- How come the experiments have the same lengths for different radii, but different lengths for different setups?

-- We used the same geographical positions for different radii because it makes the causality clear. If we used different positions for different radii, it would be unclear what makes a difference for density, wealth inequality, and so on; connection radii or subjects' geographical positions, or both. Since this study focuses on the impact of technological change (i.e., connection radii) on individual and group wealth, we use the same setup in terms of geographical positions for different connection radii. Also, the setup allows us to use paired t-test in the analysis which has better statistical power than unpaired t-test.

If R3 is referring to the length of the ties on the social network graphs (in the Figures), those were defined by the drawing algorithm and relates to the 'energy minimization' function embedded within the drawing algorithm, meant to make the images visually appealing in 2D renderings. To avoid such confusion, we remapped the graphs with geographical positions in Supplementary Figure S2.

- Supplementary Figure S3:

What is the definition of the wasted resources. Total amount which is not allocated? Somebody actually does not allocate all of their resources? Did you check who are the people who waste their resources? Could it be that those are the people who have only one neighbor and feel like that neighbor is not giving them enough? In that case it make sense that with the increase of the network the resources are less wasted, since it is less likely that nodes have only one neighbor.

Talking about that, did you analyze more deeply the behavior of the people who can have only one neighbor and who have more neighbors.

-- As R3 pointed out, we defined "wasted resource" as the total amounts that subjects did not allocate. We have explained it in the main text at page 6. We also added the definition of wasted resources in the legend of Supplementary Figure S3.

We thank R3 for suggesting the comparison of wasted resources by network degree. We added the comparison result in Supplementary Figure S3. As R3 hypothesized, subjects having only one neighbor were more likely to waste resources.

We have also showed that one-degree subjects display a lower level of reciprocity in their exchanges (see Supplementary Figure S5), and receive less resources (see Figure 5D). These results suggest that one-degree subjects have a significant disadvantage in such sharing networks. However, as we pointed out

in this paper, their disadvantage depends not only on their degree, but also on their neighbors' degree. For example, when two subjects connect only with each other, they are likely to reach mutual exchange without wasting their resources even though each subject has only one neighbor. This is one more reason motivating the introduction of the "sharing centrality" metric. Surely one-degree subjects are very likely to be placed at a disadvantaged position (even at best), but clearly, the network degree information is not sufficient to capture the whole picture. In fact, sharing centrality is more strongly correlated with the accumulated wealth than network degree (Figure 5).

- Supplementary Figure S4: The legend would make understanding of the figure much easier.

-- We updated the legend of Supplementary Figure S4 so as to focus on what the figure indicates.

Typo: Page in SI "realized"

-- Thank you for pointing out the typo. We fixed it.

REVIEWERS' COMMENTS:

Reviewer #3 (Remarks to the Author):

I am satisfied with the current version of the manuscript. It can be published as it is.